# Improving Adolescent Psychosocial Assessment through Standardized Patient Simulation: An Interdisciplinary Quality Improvement Initiative

**DOI:** 10.3390/ijerph21030283

**Published:** 2024-02-29

**Authors:** Laura Monahan, Colleen L. Eaves, Joshua C. Watson, Jordi Friese, Lisa McKenna, Erika Estrada-Ibarra

**Affiliations:** 1Department of Population Health Nursing Science, College of Nursing, Rockford Campus, University of Illinois Chicago, 1601 Parkview Avenue, Rockford, IL 61107, USA; 2Driscoll Children’s Hospital, 3533 S Alameda Street, Corpus Christi, TX 78411, USA; colleen.eaves@dchstx.org (C.L.E.); lisa.mckenna@dchstx.org (L.M.); 3Department of Counseling and Educational Psychology, College of Education, Texas A&M University, 6300 Ocean Drive, Corpus Christi, TX 78412, USA; joshua.watson@tamucc.edu; 4Nationwide Children’s Hospital, 700 Children’s Drive, Columbus, OH 43205, USA; jordi.friese@nationwidechildrens.org; 5School of Nursing, John Hopkins University, 525 N. Wolfe Street, Baltimore, MD 21205, USA; eestrad2@jh.edu

**Keywords:** adolescent, suicide, interdisciplinary, interprofessional education, clinical education strategies, patient simulations

## Abstract

Adolescent suicide and mental illness have increased at alarming rates. Healthcare professionals report a lack of skill and confidence in obtaining adolescent histories and managing confidential care due to limited training in residency. Nursing professional development practitioners face challenges of adequately preparing interdisciplinary healthcare providers to assess, identify, and intervene at all points of contact with adolescents. To increase the confidence in clinical communication skills and clinical competency, and to increase the number of social work referrals related to modifiable risk factors for adolescent patients, a Texas pediatric tertiary care center utilized standardized patient (SP) methodology to supplement traditional clinical experiences with communication-focused education based on the Home, Education, Eating, Activities, Drugs, Sexuality, Suicidality, and Safety (HEEADSSS) interviewing. This quality improvement (QI) pilot demonstrated the benefits of utilizing standardized patient methodology in communication-focused education based on the HEEADSSS interviewing. Following the SP simulations, confidence in clinical communication skills increased by 13%, clinical competency in performing comprehensive psychosocial interviews increased by 11%, use of HEEADSSS increased by 64%, and social work referrals increased by 89%. This interdisciplinary SP interviewing simulation pilot was beneficial in improving the 36 physician and nursing residents’ ability to conduct psychosocial assessments for risk factors of suicidality among adolescents.

## 1. Introduction

Despite increased societal awareness, suicide continues to plague our nation’s youth and has led to a public health crisis [1]. Intentional self-harm persists as the second leading cause of death among adolescents aged 11–21, and approximately one in five (18.8%) high school youths report considering attempting suicide [2]. In 2020, adolescent mental-health-related emergency department admissions increased by 31%, with suspected suicide attempts rising by 50.6% among females and 3.7% among males [3]. In addition to the profound emotional toll on families and communities, adolescent suicide also constitutes a heavy economic burden on society. For each completed adolescent suicide, the average total cost ranges from USD 1.2 to USD 1.4 million [4], with a present estimated overall value of total lost earning potential of USD 4.26 billion [5].

While considered a multifaceted phenomenon, mental illness, substance use disorders, previous suicide attempts, interpersonal loss, feelings of isolation or hopelessness, family factors, childhood trauma, suicide contagion, and availability of means have all been identified as risk factors contributing to adolescent suicidality [6,7]. Because primary and secondary prevention relies on an early recognition of these modifiable risk factors, the American Academy of Pediatrics recommends screening adolescents during routine history procedures [7,8]. Written instruments have proven to be useful, but some researchers suggest that up to 28% of at-risk adolescents may be overlooked when clinicians rely solely on tools such as the Patient Health Questionnaire [7]. To address this screening gap, another recommended approach is conducting a psychosocial interview through the application of the Home, Education, Employment, Eating, Activities/screens, Drugs, Sexuality, Suicidal ideation, and Safety (HEEADSSS) framework [9]. This framework assists clinicians in building rapport by moving from relatively benign conversations to more sensitive topics and can take approximately an hour to complete [10]. Unfortunately, many pediatric healthcare providers report feeling poorly equipped to interact effectively with adolescents due to a lack of training and competency [11]. When surveyed, 53% of the physician and nursing residents at a Texas 191-bed pediatric tertiary care center where this quality improvement (QI) initiative took place reported that they were unsure of their adolescent screening skills or had never had prior education regarding the HEEADSSS framework.

In 2009, the National Research Council and Institute of Medicine’s Committee on Adolescent Health Care Services and Models of Care for Treatment, Prevention, and Healthy Development [12] reported that many healthcare providers lacked the necessary skills and competencies to interact effectively with adolescents. Subsequent studies suggest little progress despite increased awareness of the deficiency. For instance, in a recent study, 32 pediatric trainees reported decreased confidence in working with adolescents in clinical practice as compared to other age groups [13]. Additionally, a survey of 53 pediatric rheumatology fellows revealed that although 61% agreed that a full comprehensive HEEADSSS assessment should be completed with all adolescent patients seen in the clinic, only 38% reported conducting a complete evaluation [14]. Accordingly, this lack of training and competency and provider discomfort were significant contributing factors to this disparity [13,14].

A promising initiative is the implementation of communication-based simulation with standardized patients (SPs) [15]. An SP is a person who has been carefully coached to simulate an actual patient’s history, body language, physical findings, and emotional and personality characteristics [16]. A study determined in two-group comparisons of 23 residents who had previously participated in structured SP adolescent training that they received a higher mean total-item score on the Structured Communication Adolescent Guide (SCAG) (*M* = 40.78) as compared to the 29 residents who had not (*M* = 32.41), *p* = 0.001 [17]. Additionally, in an experimental, two-group post-test design, another study found that 18 nursing students who participated in a 100 min SP simulation portraying a suicidal patient demonstrated higher scores in self-confidence in learning and student satisfaction compared to the 16 students who watched a recorded lecture alone, *t* (17) = −6.20, *p* ≤ 0.012 [18].

The aims of this interdisciplinary QI project were to enhance the physician and nursing residents’ interviewing processes with their adolescent patients and to improve the identification of modifiable risk outcomes within this specific pediatric tertiary care center. The purpose of this interdisciplinary QI project was to pilot an SP program based on the HEEADSSS interviewing process within the physician and nursing residencies of a pediatric tertiary care center, and to improve the early identification of modifiable risk factors of suicidality among adolescent patients.

## 2. Materials and Methods

### 2.1. Design

Aims for this QI pilot program were developed based on the concept of knowledge transfer, adoption of behaviors, and patient outcomes, which are described in the revised Ottawa Model of Research Use [19]. The Ottawa Model of Research Use is a planned action theory, which views the adoption of innovation as a purposely planned process that guides knowledge translation into practice. The framework’s key constructs (evidence-based innovation, potential adopters, practice environment, implementation strategies, adoption, and outcomes) are interconnected through the processes of the assessment of barriers and supports, are monitored throughout the implementation, and evaluate the outcomes [19]. With an assessment of the potential barriers for this pilot project, mitigation strategies were developed to include professional development and mentoring opportunities for the simulation staff, the integration of the project into the current simulation schedules, and the leveraging of volunteer services. These purposes also included assessing the potential improved confidence scores in clinical communication, improved clinical competence in performing comprehensive psychosocial interviews, and an increase in the number of social work referrals related to modifiable risk factors for patients presenting with a non-mental-health-related chief complaint following the last simulation session.

### 2.2. Sample

The sample began with forty-five physician and nursing residents at the pediatric tertiary care center, which equated to 61% of the total available group. Of these, 9 were excluded for not completing all the activities required for the data collection, resulting in a final sample size of 36 (physician residents *n* = 22, nursing residents *n* = 14). This satisfied the minimum sample requirements per power analysis. Participants’ ages ranged from 22 to 52 years (*M* = 31.9, *SD* = 7.0) and they identified as female (*n* = 23, 64%) or male (*n* = 13, 36%). Nurses reported their educational preparation as an Associates Degree of Nursing (*n* = 9, 64%) or Bachelor of Science in Nursing (*n* = 5, 36%), and physicians reported educational preparation as a Doctor of Medicine (*n* = 21, 95%) or Doctor of Osteopathic Medicine (*n* = 1, 5%). Informed consent was obtained from all participants involved in this study. A further delineation of demographics per discipline is noted in Table 1.

The project was reviewed by an Institutional Review Board and was determined to not meet the definition of research involving human subjects. Additional measures to protect the rights and confidentiality of both resident participants and SPs included a confidentiality and video consent agreement and an opt-out clause, without explanation, for any simulated scenario that made the SP or resident uncomfortable.

### 2.3. Instrumentation

The Self-Efficacy 12 (SE-12) [20] was used to evaluate resident self-confidence in communication skills. The SE-12 questionnaire consists of 12 items assessing general clinical communication based on the Calgary–Cambridge Guide [20]. Responses were recorded on a 10-point response scale ranging from 1 (very uncertain) to 10 (very certain). An average score for all 12 items was determined for each participant, with a high score indicating a higher level of confidence.

The Structured Communication Adolescent Guide (SCAG) [21] was utilized by SPs to evaluate the resident participants’ ability to perform a comprehensive psychosocial interview. SCAG, which is presented in Nelson’s Essentials of Pediatrics [22], is a 33-item checklist, based on the HEEADSSS format. The SCAG’s 33 items are composed of four sections: introduction, gathering information, teen alone, and wrap-up. Each of SCAG’s 33 items on the checklist could be given a score of 0 (did not ask), 1 (asked), or 2 (asked well) for a maximum total-item score of 66, with a higher score indicating greater competence.

Each of these tools, the SE-12 and the SCAG, are considered reliable and are validated within the literature [21,23,24].

### 2.4. Procedures

This QI pilot was conducted at a freestanding pediatric tertiary care center in a full-service, 191-bed teaching hospital that featured diverse physician and nursing residency programs. The core multidisciplinary clinical project team leaders consisted of a nursing professional development specialist, a licensed clinical social worker (LCSW), a second-year physician resident, and a nursing student. A pool of adolescent SPs was established through collaboration with a local university and the organization’s volunteer services department. All SPs were over the age of 18 and were volunteers. To prepare, SPs portraying adolescents attended a four-hour training workshop utilizing the Association of Standardized Patient Educators (ASPE) Standards of Best Practice [23]. Guardian SP roles were filled by simulation center faculty who were prepared for their presentations prior to the start of the activity. The three-hour adolescent simulation activities were scheduled every other week over three months beginning in February 2022. The twelve session participants were divided into two groups of six and assigned staggered start times approximately 40 min apart. Physician and nursing resident participants were scheduled by their respective coordinators, and attendance was highly encouraged rather than compulsory.

To prepare resident participants for their scheduled session, prerequisite education was emailed one week preceding the event with instructions to review prior to their arrival at the session. The 45 min self-study module included links to a one-page HEEADSSS guide with examples of open-ended questions, a PowerPoint detailing laws and guidelines for adolescent confidentiality in Texas, a model adolescent psychosocial interview video demonstration, and a short article regarding conducting a HEEADSSS assessment in 15 min.

Here is a brief overview of the flow of the QI pilot program: 1. Participants are briefed upon arrival and are assigned as pairs (such as a nurse–physician pair). 2. These pairs are assigned to an exam room with a standardized-patient-adolescent–guardian duo to conduct a full psychosocial interview. 3. These pairs are provided a copy of their individual results and reconvene for a forty-minute group reflection guided by the LCSW. 4. These pairs are then assigned to a second exam room with another SP-adolescent–guardian duo to conduct a second full psychosocial interview. 5. These pairs are next provided a copy of their individual results and reconvene for another forty-minute group reflection guided by the LCSW. The following paragraph gives more information on the flow of the pilot program.

Participants were briefed upon arrival at the simulation center to establish psychological safety and expectations for the encounter. The sample group of 36 physician residents and nursing residents was then divided into nurse–physician pairs, or if no nurse was available, a physician intern in their first year was paired with a senior physician resident. Each member of the pair was assigned two fictitious patients and provided a brief history of present illness for review prior to entering the exam room.

Participants were randomly assigned to an exam room where an adolescent–guardian SP duo awaited them. The fictitious patient “Taylor,” portrayed simultaneously by all three SP adolescents, was the first on the schedule. The physician or nursing resident assigned to that patient was allotted ten minutes to conduct a full psychosocial interview using any resources necessary while the second resident observed. Following the ten-minute interview, SPs were given three minutes to individually evaluate the encounter and seven minutes to engage in constructive verbal feedback utilizing the SCAG as a guide. Guardian SPs allowed the adolescent SPs to guide the debrief and supplemented when appropriate. After the individual debrief, the process was repeated for the second resident in the pair with the next fictitious patient on the schedule, “Jaime”.

Following the initial encounter, all three physician–nurse or physician–physician-resident pairs were provided a copy of their individual SCAG results and reconvened as a large group in the simulation classroom for a forty-minute group reflection guided by the LCSW. To maximize resources and provide ample opportunity for participation, during this time, the second group of six residents initiated their initial interviews.

After a group guided reflection, the physician–nurse or physician–physician-resident pairs were escorted to new exam rooms where a different set of SPs awaited them. The interview and debrief processes were repeated with the next two fictitious patients on the schedule, “Chris” and “Kelly”. This follow-up strategy was utilized to preserve the sense of unfamiliarity one might feel meeting an adolescent patient for the first time. After their two-hour simulation activity, each physician or nursing resident had directly performed two and observed two adolescent psychosocial interviews with SP feedback for a total of four encounters.

### 2.5. Analysis

Data collection commenced with the baseline survey to gather demographics, preliminary SE-12 scores, and current use of HEEADSSS in clinical practice. Response options regarding the current utilization of HEEADSSS in practice were based on a five-point Likert scale (1 = Never, 2 = Rarely, 3 = Occasionally, 4 = Frequently, and 5 = Almost Always). Subsequent SE-12 surveys were administered immediately following the activity and again four weeks after the completion of all simulation sessions. The four-week follow-up survey also asked participants to self-report using the HEEADSSS assessment in clinical practice. All surveys were distributed utilizing Qualtrics XM (Silver Lake, Menlo Park, CA, USA) experience management software. Additionally, adolescent and guardian SPs evaluated residents’ performance in the initial and subsequent interviews on a copy of the SCAG, which was later transcribed to a spreadsheet. Finally, reports of social work referrals from 1 August 2021 to 22 May 2022 were obtained from the organization’s decision support team.

Individual scores and responses were matched via the participants’ unique identification numbers and analyzed utilizing JASP 0.18.2 (Amsterdam, Netherlands) statistical software. Participants’ initial and follow-up total-item scores were aggregated, and a one-tailed paired *t*-test was performed to establish the existence of a positive change in performance for both SE-12 and SCAG scores. Results for HEEADSSS use in practice were codified where 1 = never or rarely, 2 = occasionally, and 3 = Frequently or Almost Always, and a Wilcoxon signed-rank test was used to determine significant changes in clinical use.

Social work referrals were filtered by the provider type, patient age, chief complaint, and referral reason to quantify the weekly rate of mental or behavioral health-related referrals to the social work department for adolescent patients (11–21 years) presenting with a non-mental-health-related chief complaint. The mean number of weekly referrals was analyzed utilizing a control chart. The upper and lower control limits were determined using baseline data prior to the week of implementation. The remaining weekly data following implementation were plotted on the control chart and analyzed for any special cause patterns or shifts in the center line.

## 3. Results

Significant improvements were noted in this QI pilot. These included improved confidence in clinical communication, improved clinical competence in performing comprehensive psychosocial interviews, improvements in the early identification of modifiable risk factors of suicidality among adolescent patients, and an increase in the number of social work referrals related to modifiable risk factors for patients presenting with a non-mental-health-related chief complaint following the last simulation session.

Self-confidence in adolescent communication improved by 13%. These results immediately following the SP simulation showed (*M* = 8.45), *t* (35) = −4.29, and *p* ≤ 0.001 and at four-week follow-up, (*M* = 8.64), *t* (10) = −5.94, and *p* ≤ 0.001 as compared to baseline (*M* = 7.68). For the clinical competence in performing comprehensive psychosocial interviews, the pre- and post-SCAG results revealed improvements in participants’ performance of the SCAG 33-item checklist from the initial (*M* = 50.42) to the follow-up interview (*M* = 55.83), *t* (35) = −3.37, and *p* ≤ 0.001, and an overall 11% increase in self-efficacy in communicating with adolescents from baseline. Each SCAG section did have the option to assign a global score (A, B, C, D, F); however, this was not completed consistently and thus was not utilized in the analysis. Additionally, improvements in the early identification of modifiable risk factors of suicidality among adolescent patients were demonstrated by the incidence of residents self-reporting the use of HEEADSSS in clinical practice, as either Frequently or Almost Always designations increased from 47% at baseline to 64% at four-week follow-up. A Wilcoxon Signed-Rank Test indicated a significant mean rank change from baseline (*M* = 2.16) to follow-up (*M* = 2.55), *p* = 0.02.

Finally, five data points rose above the upper control limit following program implementation as noted in Figure 1. This denoted a shift in the mean weekly referrals from (*M* = 1.22) to (*M* = 2.31), indicative of an increase in the identification of adolescents who could benefit from speaking with a social worker related to potentially modifiable risk factors of suicidality. In analyzing the control charts at this Texas pediatric tertiary clinical care center, these findings indicated an 89% increase in the number of social work referrals that were made related to modifiable risk factors for patients presenting with a non-mental-health-related chief complaint.

## 4. Discussion

In a recent systematic review, researchers agreed that SP-based education was accepted as a valuable means of improving the clinical communication and clinical competency skill sets of healthcare providers [25]. Nonetheless, the implementation of standardized patient methodology can be notably resource-intensive [26,27], so the intent of this pilot was to evaluate the clinical value of this type of activity in the local setting prior to introducing the program formally into the physician and nursing residency curriculums. Researchers report a positive correlation between an increase in self-efficacy and improved skill performance related to simulation training [28]. In this project, it was noted that participants not only sustained an immediate 10% increase in self-confidence scores but gained an additional 3% in subsequent scores. This was promising, suggesting that the simulation activity contributed synergistically to the self-confidence achieved through traditional clinical experiences.

Other researchers also note that effective communication promotes a patient’s trust in a healthcare provider and increases the likelihood that they will communicate private and personal information more freely [29]. The results of this pilot indicate a significant 11% improvement in performance of the psychosocial interview in-between the initial and follow-up encounters, as evaluated by the adolescent SPs. During the activity, residents were allowed to utilize resources to conduct the psychosocial interview, and many opted to bring notes into the exam room. Therefore, even if they had limited experience with the HEEADSSS assessment, residents scored relatively well on the initial and subsequent encounters.

Another significant finding was the increase in residents’ self-reported performance of a full HEEADSSS assessment with adolescent patients through the designations of either Frequently or Almost Always by 17%. Similarly, another study noted in their adolescent emergency medicine SP simulation quality improvement project that at the three-month follow-up, participants indicated an increased use of new skills in practice, especially through the use of the HEEADSSS assessment tool [30]. Other researchers found that individuals were less likely to present with future self-harm injuries following participation in psychosocial assessments [31]; therefore, the increase in the frequency of HEEADSSS assessments was beneficial to the adolescent populations.

A study that reviewed sixty adolescent charts found that those presenting for other health conditions had fewer completed HEEADSSS assessments on file compared to those with mental health presentations, 3% compared to 47%, respectively [32]. However, the Substance Abuse and Mental Health Services Administration [33] acknowledges that individuals with experiences of adverse childhood events and trauma are found in multiple service sectors, not just in behavioral health. In analyzing the control charts at this Texas pediatric tertiary clinical care center, findings indicated an 89% increase in social work referrals for these types of patients. The assumption was that an increase in social work referrals would indicate an increase in the identification of modifiable risk factors for suicidality. This is a noteworthy outcome and significant to the body of SP quality improvement literature as most studies focus on reactions and learning rather than clinical behaviors and patient outcomes.

This SP interviewing simulation QI pilot demonstrates various strengths and implications for clinical assessment of adolescent mental health issues. Standard patient simulations, such as the one utilized in this QI clinical education project, can effectively enhance nursing preparation and practice by delivering a more seamless transition in the development of competencies for nursing residents, especially when accompanied by the inclusion of patient care interactions with other interdisciplinary team members. Notably, SP programs, especially those using interprofessional collaboration, can be utilized by all healthcare disciplines to develop and refine motivational interviewing skills, enhance providers’ practice of collaboratively managing difficult behaviors, and culturally prepare providers for successfully engaging in challenging or problematic conversations. This is particularly relevant in increasing various healthcare providers’ competency developments in their communications and interactions with at-risk adolescents.

This interdisciplinary SP interviewing simulation QI pilot was beneficial in improving the 36 nursing and physician residents’ ability to conduct psychosocial assessments for risk factors of suicidality among adolescents. An increased transfer of knowledge was noted through the enhancement of the nursing and physician providers’ confidence and competence within this project’s simulation lab, as the concept of adoption corresponded to the increased reported use of HEEADSSS in practice, and the outcomes were directly related to the number of adolescents identified who could benefit from additional social work support. This is notable that by improving the effective communication skills of the nursing and physician residents, a more thorough screening was performed to ensure that those at-risk adolescents who might benefit from a social work consult would receive it in an efficient and timely manner. This SP simulation QI pilot will be a valuable addition to both the nursing and physician residency programs, especially since much of the cost associated with the planning, design, and development of the curriculum has already been completed. Moving forward, the next phase in this QI plan will be to integrate the adolescent SP psychosocial interprofessional collaborative assessment activity into the academic schedule to coincide with residents’ adolescent and developmental health rotations.

Our community’s youth are experiencing a mental health crisis, culminating in rising adolescent suicide rates. An early identification of modifiable risk factors can help adolescents receive the support they need to successfully navigate this vulnerable stage of growth and development. An early identification of those modifiable risk factors could also potentially decrease the number of suicide rates among this age group, especially among the female adolescent population who are most at risk.

While this QI pilot had many strengths, there were some limitations. One of the limitations was the inability to accurately measure social work requests originating from nursing as all screenings are documented under the admitting physician. An additional limitation was that no formal measure was established to consider the effects of the interdisciplinary aspect of the activity, although one nursing participant stated “This is so helpful to hear how the doctors interact with the patients and what types of questions they ask”. Additional limitations were that there was a potential for biases in self-report measures for the increases in confidence and self-efficacy, and as this was a pilot program, there was no control group or comparison with other training methods, or with traditional training approaches regarding effectiveness and efficiency. The authors acknowledge that the lack of a control group contributes to the limited referential nature of the findings and conclusions in this QI pilot project.

## 5. Conclusions

Subsequent to the adolescent SP simulations, confidence in clinical communication skills of the nursing and physician residents increased by 13%, clinical competency in performing comprehensive psychosocial interviews increased by 11%, use of HEEADSSS increased by 64%, and social work referrals increased by 89%. Although many pediatric healthcare providers voice a lack in confidence in their ability to interact effectively with adolescents, the literature supports the use of adolescent SP simulation as an educational modality to improve clinicians’ interactions with adolescents. The result of this QI pilot indicates the strength of this recommendation as well. With the current emphasis on improving nursing education’s assessment and evaluation methods through the enhancement in teaching modalities, especially those incorporating interprofessional education, simulation, and collaboration, this QI pilot exhibits advancements within these associated realms.

In the future, the adolescent SP interviewing simulations will become a routine course of the interdisciplinary interactions offered by this pediatric tertiary care center’s simulation center. The success of this QI pilot warrants an ongoing re-evaluation of this particular organization’s previously siloed educational model to assess whether additional opportunities for incorporating and encouraging interprofessional education to promote the mutual appreciation of interdisciplinary collaboration can be further achieved. Thus, other healthcare providers and disciplines will be invited to take part in this program to help them consider how SP methodology can be used to improve the quality of care offered in their designated specialty areas.

Additionally, aside from the positive interprofessional collaborative developments exhibited in this experience itself, participants’ use of written resources to guide the adolescent interview processes had improved their performance significantly. Currently, there are no specific cues within the EMR for the HEEADSSS framework, which may indicate another opportunity for future improvements. Due to the intricacies of Texas confidentiality laws, this will require close collaboration with the health records, ethics, and compliance offices to ensure positive outcomes for healthcare providers, patients, and guardians. The development of closer collaborative efforts evidenced through this project can augment those continuing developments to achieve these results.

The results of this QI pilot manifested positive steps in the enhancement in the training of the nursing and physician resident interprofessional teams and increased their educational initiatives for collaborative competencies and interactions. This pilot demonstrated improvements in the interdisciplinary healthcare providers’ practices to assess, identify, and intervene in their points of contact with at-risk adolescents. These confidence and competency improvements demonstrated by this QI pilot for nursing and physician residents to conduct adolescents’ psychosocial assessments for risk factors of suicidality can further enhance adolescents’ ongoing care. It is a win/win outcome for all.

## Figures and Tables

**Figure 1 ijerph-21-00283-f001:**
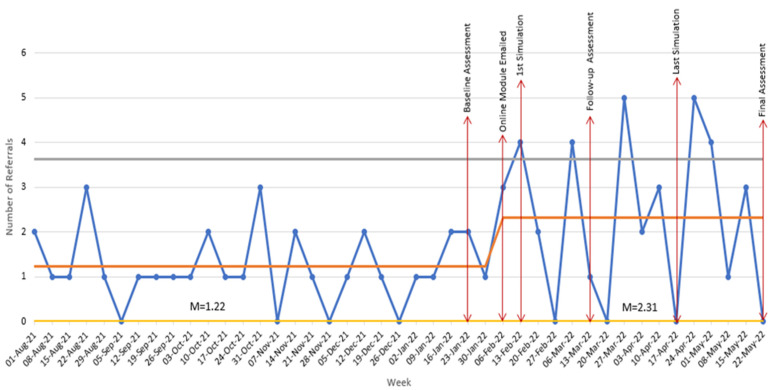
Weekly physician resident referrals to social work for adolescents presenting with non-mental or behavioral health primary diagnosis. The Yellow Line is the lower control limit of the number of psycho-social referrals to social work; The Orange Line is the mean average of the number of psycho-social referrals to social work; The Blue Line is the actual number of psycho-social referrals to social work; The Grey Line is the upper control limit of the number of psycho-social referrals to social work.

**Table 1 ijerph-21-00283-t001:** Participants’ Demographics.

	Physician	Nursing	Total
Number of Residents	22	14	36
Age			
Mean (*SD*)	34.9 (6.7)	27.3 (4.4)	31.9 (7.0)
Gender			
Female	12 (55%)	11 (79%)	23 (64%)
Male	10 (45%)	3 (21%)	13 (36%)
Ethnicity			
Asian	5 (23%)	1 (7%)	6 (17%)
Black or African American	8 (36%)	0 (0%)	8 (22%)
Hispanic, Latino, or Spanish Origin	5 (23%)	7 (50%)	12 (33%)
White	3 (14%)	2 (14%)	5 (14%)
Two or More Ethnicities or Other	1 (4%)	4 (29%)	5 (14%)
Education			
Doctor of Medicine (MD)	21 (95%)		
Doctor of Osteopathic Medicine (DO)	1 (5%)		
Associate Degree in Nursing (ADN)		9 (64%)	
Bachelor of Science in Nursing (BSN)		5 (36%)	
Previous Yrs. of Healthcare Experience			
None	7 (32%)	7 (50%)	14 (39%)
0–3 Years	4 (18%)	3 (21%)	7 (19%)
4–9 Years	7 (32%)	2 (14%)	9 (25%)
10+ Years	4 (18%)	2 (14%)	6 (17%)
History of HEEADSSS Education			
No	6 (27%)	8 (57%)	14 (39%)
Unsure	1 (5%)	4 (29%)	5 (14%)
Yes	15 (68%)	2 (14%)	17 (47%)
Physician Post-Graduate Year			
PGY-1	11 (50%)		
PGY-2	7 (32%)		
PGY-3	4 (18%)		
Time in Nurse Residency			
0–2 Months		8 (57%)	
3–5 Months		3 (21%)	
6–8 Months		3 (21%)	

## Data Availability

Data are available on request due to restrictions, e.g., privacy.

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
