# Peer review of "Improving Adolescent Psychosocial Assessment through Standardized Patient Simulation: An Interdisciplinary Quality Improvement Initiative"

_ijerph, 2024, doi:10.3390/ijerph21030283_

Round 1
Reviewer 1 Report
Comments and Suggestions for Authors
Title: Improving Adolescent Psychosocial Assessment through 2 Standardized Patient Simulation: An Interdisciplinary Quality 3 Improvement Initiative
An interesting study; the manuscript is well-written - I have very few comments.
Abstract:
Suggest that the authors define SP when first introduced in the abstract. SP first appears as follows: “following the SP simulations”
Introduction
1. A comprehensive literature review.
2. Either here or some place in the manuscript please indicate how long it takes to administer the HEEADSSS.
3. Typo in 3rd paragraph – it reads HEADSSS instead of HEEADSSS
4. Last paragraph of introduction – are there any other relevant studies that could be included?
Materials and Methods
1. Are there additional details about the Ottawa Model of Research Use that would be useful to include?
Comments on the Quality of English Language
English language is fine - well-written.
Author Response
Responses to Reviewer 1
Thank you very much for taking the time to review this manuscript. We appreciate your comments. Please find the detailed responses below and the corresponding revisions/corrections highlighted/in track changes in the re-submitted files.
Suggest that the authors define SP when first introduced in the abstract. SP first appears as follows: “following the SP simulations”
Response: Addressed in the Abstract: standardized patient (SP)
Introduction
- A comprehensive literature review. Response 1: Thank you.
- Either here or some place in the manuscript please indicate how long it takes to administer the HEEADSSS. Response 2: The Home, Education, Employment, Eating, Activities/Screens, Drugs, Sexuality, Suicidal ideation, and Safety (HEEADSSS) framework assists clinicians in building rapport by moving from relatively benign conversations to more sensitive topics and can take approximately an hour to complete.
- Typo in 3rdparagraph – it reads HEADSSS instead of HEEADSSS Response 3: Corrected.
- Last paragraph of introduction – are there any other relevant studies that could be included? Response 4: Relevant studies are already listed.
Materials and Methods
- Are there additional details about the Ottawa Model of Research Use that would be useful to include? Response 5: Added additional information on the Ottawa Model of Research Use.
Reviewer 2 Report
Comments and Suggestions for Authors
The paper addresses a critical and timely issue: the increase in adolescent suicide and mental illness. The research involves physicians and nursing residents, emphasizing the importance of interdisciplinary education in healthcare. However, The problem with this study is that it is rather premature, inferential, and observational.
Methodological Concerns
- Limited generalizability due to the study being conducted at a single pediatric tertiary care center.
- The potential for biases in self-report measures, such as empirical and temporal increases in confidence and self-efficacy.
- Absence of a control group or comparison with other training methods.
- No discussion on how this method compares with traditional training approaches regarding effectiveness and efficiency.
Overall, the data are poorly articulated, and the manuscript is difficult to read. Data are not presented in a logical order and therefore, are difficult to follow.
Potential for Improvement
- "2. Materials and Methods" should be written more clearly. It is unclear whether it is about the design, the purpose, the procedure, or the flow of the study. For example, how about illustrating the methods in chronological order?
- The primary and secondary endpoints should be organized. For instance, the statement "five data points rose above the upper control limit following program implementation as noted in Figure 1." has no scientific meaning.
- If comparing averages, appropriate figures and tables should be used, accompanied by thorough explanations.
Author Response
Responses to Reviewer 2
Thank you very much for taking the time to review this manuscript. We appreciate your comments. Please find the detailed responses below and the corresponding revisions/corrections highlighted/in track changes in the re-submitted files.
The paper addresses a critical and timely issue: the increase in adolescent suicide and mental illness. The research involves physicians and nursing residents, emphasizing the importance of interdisciplinary education in healthcare. However, The problem with this study is that it is rather premature, inferential, and observational.
Methodological Concerns
- Limited generalizability due to the study being conducted at a single pediatric tertiary care center.
Response 1: This has been addressed in the Limitations area.
- The potential for biases in self-report measures, such as empirical and temporal increases in confidence and self-efficacy.
Response 2: This has been addressed in the Limitations area.
- Absence of a control group or comparison with other training methods.
Response 3: This has been addressed in the Limitations area.
- No discussion on how this method compares with traditional training approaches regarding effectiveness and efficiency.
Response 4: This has been addressed in the Limitations area.
Overall, the data are poorly articulated, and the manuscript is difficult to read. Data are not presented in a logical order and therefore, are difficult to follow.
Response 5: This has been reconfigured to address this request.
Potential for Improvement
- "2. Materials and Methods" should be written more clearly. It is unclear whether it is about the design, the purpose, the procedure, or the flow of the study. For example, how about illustrating the methods in chronological order?
Response 6: An overview of the chronology has been addressed in Material and Methods section.
- The primary and secondary endpoints should be organized. For instance, the statement "five data points rose above the upper control limit following program implementation as noted in Figure 1." has no scientific meaning.
Response 7: This has been reconfigured to address this request.
- If comparing averages, appropriate figures and tables should be used, accompanied by thorough explanations.
Response 8: This has been configured and addressed as a QI pilot program.
Round 2
Reviewer 2 Report
Comments and Suggestions for Authors
This study faces significant reproducibility issues due to the lack of a control group and insufficient description of the SP simulation, which still need to be resolved despite revisions. Nonetheless, it tackles critical issues related to adolescent health behaviors and engagement, representing a meaningful effort in public health. Considering this research is a pilot program, its future development is anticipated with interest, suggesting a foundation for more comprehensive future studies that could address these methodological concerns.
Author Response
Thank you very much for taking the time to review this manuscript. We appreciate your comments. Please find the detailed responses below and the corresponding revisions/corrections highlighted/in track changes in the re-submitted files.
The lack of a control group is listed as a limitation of the pilot project.